# CaptainA - A mobile app for practising Finnish pronunciation

**Nhan Phan**
Aalto University
nhan.phan@aalto.fi

**Tamas Grosz**
Aalto University
tamas.grosz@aalto.fi

**Mikko Kurimo**
Aalto University
mikko.kurimo@aalto.fi

## Abstract

Learning a new language is often difficult, especially practising it independently. The main issue with self-study is the absence of accurate feedback from a teacher, which would enable students to learn unfamiliar languages. In recent years, with advances in Artificial Intelligence and Automatic Speech Recognition, it has become possible to build applications that can provide valuable feedback on the users' pronunciation. In this paper, we introduce the CaptainA app explicitly developed to aid students in practising their Finnish pronunciation on handheld devices. Our app is a valuable resource for immigrants who are busy with school or work, and it helps them integrate faster into society. Furthermore, by providing this service for L2 speakers and collecting their data, we can continuously improve our system and provide better aid in the future.

## 1 Introduction

Proper pronunciation is needed to build confidence in second language (L2) learners and is essential for effective communication and language acquisition (Gilakjani, 2012). L2 adult learners, who might not have regular exposure to the target language during their everyday life, may lack sufficient opportunities to practise and receive corrective feedback.

With recent advances in Automatic Speech Recognition (ASR) technologies, computer-assisted pronunciation training (CAPT) apps have become more and more effective in helping L2 learners. These apps can immediately give the users feedback on their pronunciation at their convenience. However, while popular languages such as English have many pronunciation applications (Kholis, 2021; Fouz-González, 2020;

Wellocution, 2023), there are fewer resources available for Finnish L2 learners. To the best of our knowledge, there was no similar app for CAPT in Finnish before this work.

The main challenge in developing CAPT applications for Finnish and other low-resource languages is the lack of data from L2 speakers. Furthermore, if the L2 corpus is not annotated at the phoneme level, it makes developing an app for mispronunciation detection (MD) more complicated. We designed our CaptainA app to function as well as possible using all available data and add the possibility of collecting users' data after the pilot phase (figure 1). Such information will help evaluate the app's effectiveness for language training and improve our model's performance to better address students' needs in later versions.

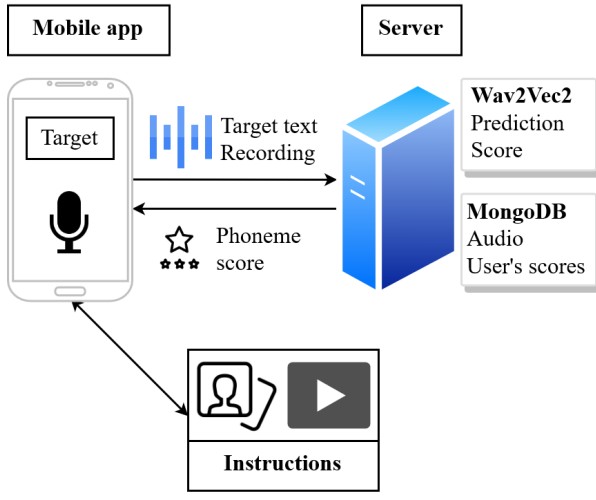

Figure 1: CaptainA app processing flowchart

Recent works from Wu et al. (2021) and Xu et al. (2021) have demonstrated the effectiveness of end-to-end systems with Transformer-based architectures for English MD. While we focus more on practicality, we use a similar approach without a detailed annotation dataset for Finnish.

## 2 Dataset

One of the major challenges that we needed to overcome was the limited data at our disposal. We should note that for the English language, several datasets are available with phoneme level annotation (Zhao et al., 2018; Zhang et al., 2021; Weinberger, 2015). Unfortunately, no such public Finnish resources exist. Thus we opted to use the data collected during the Digitala project (Al-Ghezi et al., 2023) as our primary corpus. This dataset includes ratings from language experts on pronunciation, fluency, lexical, grammatical and the holistic overall level for each audio file, but it does not have phoneme level information.

The Digitala corpus consists of free-form and read-aloud speech, from which we selected 768 short read-aloud samples as those matched our intended scenario most closely. This gave us approximately 60 minutes of audio with the overall pronunciation ratings ranging from 1 to 4, with 4 being the best. The rating is for the whole pronunciation task and not individual phonemes. The lowest pronunciation level (1) contains approximately 2,200 phones, the highest one (4) has only 576 phones, while the remaining 14,000 phones are split almost equally between levels 2 and 3. The corpus was also transcribed by third parties who were not language experts.

The small size of the Digitala corpus and the lack of phoneme annotation meant it was not suitable for training or finetuning for the MD task. However, as there were no better alternatives, we used the Digitala read-aloud transcript as a replacement for the evaluation set. Consequently, we needed another dataset to train our models. After some preliminary experiments, we selected the Finnish Parliament corpus (Kielipankki, 2022), a publicly available corpus without any statistically significant use of dialects (Virkkunen et al., 2023). By training our models for the ASR task with suitably chosen native speakers' samples, we expected the models could learn the features of native Finnish speech and have the potential to identify deviations made by L2 speakers. As a first step, we filtered the most suitable portion of the data, by selecting speeches with low or average speaking rates (which is the most similar to how L2 learners speak). As an additional step, we also restricted the data by excluding older (50+) speakers, since our target audience is generally younger immigrants. The last step in data preparation was the splitting of the 281 hours of data into 75% for training, and 25% for tuning hyperparameters and evaluating the speech recognition models. We should note that we also used two publicly available reference models, called Finnish-NLP[1] and Finnish-NLP-S [2]. Both have been trained with 228 hours of Finnish Parliament data and approximately 47 hours of data from other sources.

## 3 Implementation

### 3.1 Server

The core technology inside our server is based on wav2vec 2.0 (Baevski et al., 2020), which was already proven to work exceptionally well even with very limited amount of data (Wu et al., 2021; Xu et al., 2021). We selected XLS-R (Babu et al., 2022) and Uralic, a subset of VoxPopuli (Wang et al., 2021), as our pre-train models, and use the state-of-the-art model in Finnsh ASR, Finnish-NLP, as our baseline. Except for entropy $\beta$, all models used the same hyperparameters, and there is no language model used for decoding.

Leveraging the phonetic nature of the Finnish language, where each phoneme is represented by exactly one grapheme[3], we can use graphemes as output units during the ASR training procedure. Once the ASR models were trained, we used the forced alignment algorithm for Connectionist Temporal Classification (CTC) from Kürzinger et al. (2020) to determine the success of pronunciation. This algorithm provides both time alignment and a probability score for each grapheme. Inspired by the traditional Goodness of Pronunciation method (Witt and Young, 2000), we use such information to generate feedback for the user.

One major issue we had to overcome was the overconfidence of the wav2vec 2.0 models. As it is well known, the CTC algorithm often results in spiky outputs (Zeyer et al., 2021), which in terms would mean that we can only provide binary (correct/incorrect) feedback to the user. Naturally, a good pronunciation training app should give more detailed information (Engwall and Bälter, 2007), thus, reducing the peakedness of the outputs was important. To achieve this, we chose the negative maximum entropy regularization technique Liu

---

[1]https://huggingface.co/Finnish-NLP/wav2vec2-xlsr-1b-finnish-lm-v2

[2]https://huggingface.co/Finnish-NLP/wav2vec2-xlsr-300m-finnish-lm

[3]except "nk" [ŋk] and "ng" [ŋː]

| Model | Vocabulary | Parameters | Entropy $\beta$ | CER | Recall | Precision | $F_1$ |
|---|---|---|---|---|---|---|---|
| Finnish-NLP | Grapheme | 1bil | 0% | **15.4%** | 59.8% | **33.3%** | **42.8%** |
| Finnish-NLP-S | | 300mil | | 22.3% | **65.0%** | 26.1% | 37.2% |
| XLS-R | Grapheme | | 0% | 20.9% | 61.1% | 26.7% | 37.2% |
| XLS-R-5 | Grapheme | | 5% | **19.5%** | 63.1% | **30.0%** | **40.6%** |
| XLS-R-10 | Grapheme | 300mil | 10% | 21.2% | 63.1% | 29.4% | 40.1% |
| XLS-R-10-P | Phoneme | | 10% | 21.3% | 63.2% | 27.3% | 38.1% |
| Uralic-10 | Grapheme | | 10% | 30.4% | 64.3% | 23.4% | 34.3% |
| Uralic-10-P | Phoneme | | 10% | 29.6% | **66.8%** | 22.6% | 33.8% |

Table 1: Speech models' performance in ASR and MD on Digitala read-aloud set.

et al. (2018) during training, which redistributes $\beta$% of the total probability mass uniformly to all outputs, ensuring the smoothness of the final predictions.

## 3.2 Mobile app

We use Unity (Juliani et al., 2020) as our development engine. With Unity we can simultaneously publish our CaptainA app to multiple platforms: Android, iOS and Windows. Our app contains various study materials, and Unity Editor allows us to easily integrate those multimedia content into the app. We make use of the engine to visualize our pronunciation instructions with animations and limit the rest to simple UI, thus lowering the application's power consumption.

Arapakis et al. (2021) estimated a 7 seconds threshold where mobile (web search) users' experience decreases significantly. To maintain a reasonable response time, we use a manual VAD system to remove the silent parts from the recording: the users must press and hold the record button to record their audio samples.

The app supports two modes; the "Topic" mode supplies curated words and phrases for various topics (Easy, Normal, Hard, Greetings, Grocery, similar vowels pair, or classic Finnish literature...), often along with English translation and audio samples from native speakers. On the other hand, the "Freestyle" mode enables users to practice any word or phrase by first prompting for the text that the user will attempt to pronounce.

The score for each phoneme is saved locally, enabling users to track their progress. The data is valuable in developing speech applications for L2 speakers. In the future, with the users' permission, we can collect their records to evaluate the app's effectiveness and other metadata.

CaptainA also provides pronunciation instruc-

tions via sample audios, pictures, animations and videos, which are beneficial for users during self-practice (Engwall and Bälter, 2007). The audio, photo and animation materials are directly stored in the app, while the videos are accessible via a public, ad-free platform. We should note that external links would generally have an adverse effect on user experience, still we choose this solution to supply high-quality tutorial videos while keeping the size of the app reasonably small.

## 4 Results

To validate our models, we computed their character error rate (CER), Recall (percentage of mispronunciations correctly detected) and Precision (the ratio of detected mispronunciations actually being mispronunciation, according to a native Finnish listener) using the Digitala read-aloud corpus. The empirical results can be seen in Table 1. The first thing that we noticed is that the large Finnish-NLP produced significantly lower and the small Finnish-NLP-S higher CER compared to the majority of our models. Next, we compared the models in terms of MD and saw that Finnish-NLP yielded the highest overall $F_1$ score. However, the smaller XLS-R-5 and XLS-R-10 managed to achieve comparable results with the help of entropy regularization.

The benefit of entropy regularization is seen when we increase the value of $\beta$ and note that both Recall and Precision also increase. From our experiment, we found that $\beta$ between 5% and 10% produces the best result for MD task. Looking at the detailed breakdown in table 2, we also found that, the smaller XLS-R outperformed the Finnish-NLP in Recall for pronunciation level 1 samples, while slightly falling behind in Precision. The gap in Precision widens as the speakers' pronunciation skill improves. Considering the practicality of

| Model | CER | Recall | Precision |
|---|---|---|---|
| Finnish-NLP | **26.9%** | 72.6% | **38.7%** |
| XLS-R-5 | 31.4% | 77.4% | 36.2% |
| XLS-R-10 | 33.5% | **78.5%** | 36.8% |
| Finnish-NLP | **20.0%** | 61.5% | **32.7%** |
| XLS-R-5 | 24.1% | **63.3%** | 29.1% |
| XLS-R-10 | 24.7% | 63.2% | 28.9% |
| Finnish-NLP | **11.6%** | 42.4% | **27.2%** |
| XLS-R-5 | 15.4% | **46.7%** | 24.1% |
| XLS-R-10 | 17.6% | 45.7% | 22.2% |
| Finnish-NLP | **6.0%** | 18.8% | **20.0%** |
| XLS-R-5 | 10.3% | **25.0%** | 16.0% |
| XLS-R-10 | 13.6% | **25.0%** | 10.0% |

Table 2: CER, Recall and Precision for the pronunciation levels 1 to 4 (top to bottom: worst to best)

smaller models, they would be suitable in MD for beginner L2 learners. While the Uralic model did, in our preliminary experiment on Common Voice 7.0 test set, produce lower CER on native Finnish speakers, it failed in both ASR and MD task on L2 speakers. One possible reason is that the Uralic models were not exposed to foreign language families, unlike the XLS-R models.

One common type of mistake made by Finnish L2 speakers is related to the duration of phonemes. In our test set, excluding other types of mistakes, the XLS-R-10 model achieved approximately 68% Recall rate where users pronounce short (single) phonemes too long, and 57% where users pronounce long (double) phonemes too short.

Although our goal was to develop a CAPT app that works well for all users, due to the composition of the training data and the randomness in the optimization process, the system still developed some biases towards subgroups of users. For samples in the same pronunciation level, the XLS-R-10 model has better MD for *male* speakers, with a higher Recall rate (66.0% vs *females* 59.8%), and a slightly higher Precision (31.2% vs 27.6%). The gender distribution in the test set is balanced, with 53% *male* samples and 47% *female*. We will release more detailed analyses in the future once we collect more data from users.

While it is possible to use the training part of the Digitala corpus for finetuning our wav2vec 2.0 models, we could not control the pronunciation quality, as the speakers are L2 learners and there

is no phoneme annotation. In our preliminary experiments we found that finetuning with bad pronunciation data led to lower performance in MD.

# 5 Self-study assistant

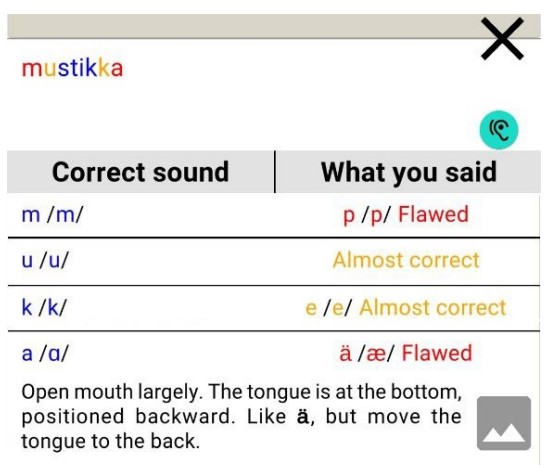

Figure 2: The result is coloured based on pronunciation score.

CaptainA (see figure 1) allows users to enter words into a text prompt to practise pronunciation. Their audio is sent to the server, and the device will display the obtained rating for each phoneme, with three possible ratings in colors (figure 2): flawed (phoneme is not recognizable), almost correct (improved, but not clear), and correct. The "almost correct" rating is given as positive feedback when user's phoneme score improves, but is still not considered correct, as suggested by Engwall and Bälter (2007).

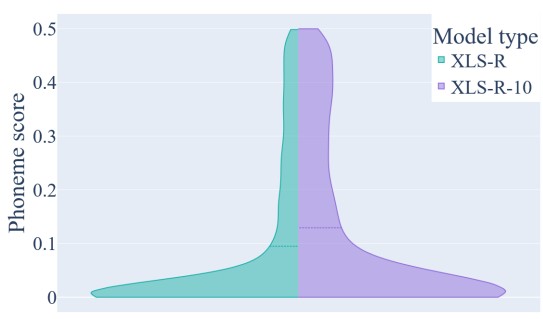

Figure 3: Phoneme score distribution for phonemes with scores less than 0.5 in the test set.

We based our ratings on the phoneme score distribution of the test set. With model XLS-R-10, any mispronounced phoneme with a score higher than 0.2 (higher than 70% percentile of

all mispronounced phonemes), or correct pronounced phoneme with a score lower than 0.8 (lower than 10% percentile of all correct pronounced phonemes) is classified as the intermediate rating "almost correct". In CAPT, we consider the cost of misclassifying correct pronunciation as incorrect more severe (Bachman et al., 1990), and therefore mispronunciation has a higher chance of being classified to a higher rating. It should be noted, we were able to introduce an additional rating category from the binary incorrect/correct dataset by implementing the entropy regularization described earlier. Higher entropy is also the practical reason to select the XLS-R-10 model instead of the XLS-R-5 for CaptainA.

When making mispronunciation, the users are also advised to refer to the app multimedia pronunciation instructions (figure 4).

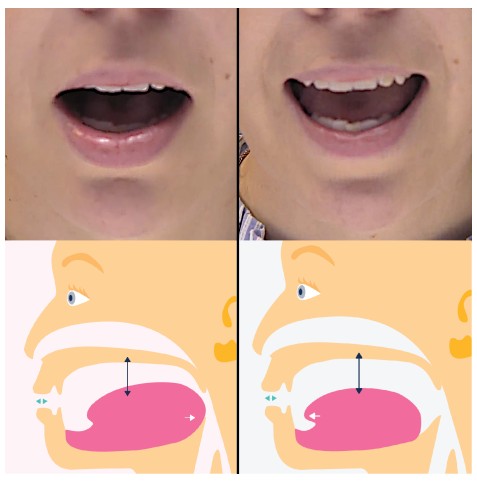

Figure 4: Visual pronunciation instructions for A [ɑ] (left) and Ä [æ] (right).

## 6 Conclusion

In this paper, we presented the prototype of CaptainA, an app that helps language learners practise Finnish pronunciation. Because of the lack of data available for phoneme level pronunciation mistakes, our solution is based on multilingual wav2vec 2.0 models, which are finetuned for native Finnish ASR. By running the L2 learners' utterances through the ASR without a language model, we predict pronunciation errors and probability scores that indicate the success of pronunciation. The resulting models are validated by measuring CER, Recall and Precision for samples of different levels of pronunciation judged by human experts. In the future, we plan to collect user data

(feedback and audio) with our app to update the models and improve the self-study application.

CaptainA was trained on a native speaker corpus and evaluated on 1 hour of transcribed L2 data, without requiring detailed annotation from experts or fine-tuning L2 data. Its performance could be a motivation to apply a similar setup to other low-resource languages. Based on the initial result of CaptainA, we plan to develop a similar CAPT app for Finland Swedish.

## Acknowledgments

The computational resources were provided by the Aalto Science-IT. CaptainA was funded by the Kielibuusti project. This work was also supported by NordForsk through the funding to Technology-enhanced foreign and second-language learning of Nordic languages, project number 103893. The video and photo instructions were made with the help of the Aalto University Language Centre. The side mouth illustrations are from Aino Huhtaniemi.

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
