# OpenReview forum: "CaptainA - A mobile app for practising Finnish pronunciation"
_NoDaLiDa/2023/Conference — NoDaLiDa 2023_

### Official Review · Reviewer_qgvq · 2023-03-10
**Review of the paper "APP NAME - A mobile app for practising Finnish pronunciation"**

**Rating:** 8
**Confidence:** 4

**Review:**

The paper introduces an application for pronunciation training in Finnish.
The core of the application is an ASR module for Finnish built on the state-of-art technology. Choosing Unity as the development platform provides the necessary flexibility to launch the app on different platforms and easily edit the content of the training exercises.
An important feature of the app is to give feedback to language learners on their pronunciation by highlighting the mispronounced segments. Users should also benefit from multimedia pronunciation guides for different sounds.
The paper is well written and easy to follow.

Remarks:

- There is a terminological error in Ch 2 - a phoneme is an abstraction of a set of speech sounds (phones) that are perceived as equivalent to each other, use the terms phones or speech segments instead of phonemes.
- In Finnish, as a quantity language, the duration contrast of short/long vowels and consonants is crucial. It might be difficult to acquire,
especially for those L2 learners whose native language phonology does not involve duration-based contrasts. The evaluation of the temporal characteristics of the pronounced sounds is not discussed in the paper.
- References are not ordered alphabetically.

**Paper Type:**

Short paper

---

### Official Review · Reviewer_FgTq · 2023-03-10
**Good paper but only partial evaluation**

**Rating:** 8
**Confidence:** 3

**Review:**

The paper outlines a mobile application designed to assist Finnish learners in refining and assessing their Finnish pronunciation skills. While technically sound and well-written, the primary emphasis of the paper is on training and evaluating Automatic Speech Recognition (ASR) models, with less attention paid to the multiple dimensions of the app itself. As a result, the paper feels somewhat lacking in completeness, particularly with regard to the assessment of the app, which is deferred to future research.

For example, it would be interesting to learn the following:

- would an approach that is based on a general purpose ASR engine (available on any mobile phone, also for Finnish), i.e. without detailed feedback on phonemes, work equally well, i.e. how useful are the multimedia clips (with position of the tongue etc.)
- what is the impact of carefully selecting the practice phrases on the overall outcome (improved pronunciation). Although the paper mentions "curated words and phrases," no specifics are given. One could imagine various selections to make the learning fun and/or overall useful for the everyday life: snippets of daily news, tongue twisters, standard phrasebook examples, well-known Finnish named entities (e.g. names of actors and politicians), phrases that include a wide variety of different phonemes, etc.
- does the app work better with of some (e.g. larger) immigrant communities, than with others

**Paper Type:**

Short paper

---

### Official Review · Reviewer_grLF · 2023-03-12
**The useful app for learning Finnish pronunciation**

**Rating:** 7
**Confidence:** 4

**Review:**

The paper deals with a development and structure of a mobile app for practising Finnish pronunciation.  The app helps language learners practice Finnish pronunciation. Due to the lack of data on phoneme-level pronunciation errors, the solution described in this article is based on multilingual wav2vec 2.0 models that are fine-tuned to native Finnish ASR. Furthermore, by providing  this app the authors of the article plan to collect data of L2 speakers and continuously improve the system and provide better aid in the future.

The proposed Visual pronunciation instructions for Finnish phonemes are also useful. They certainly help to understand the articulation of sounds, but only possible for those who have at least some knowledge of phonetics.

The paper is well written and interesting to read, however I see the some unclear questions:
What are the criteria that is used to display the obtained rating for each phoneme:  flawed, almost correct, and correct?
Is the interaction of phonemes in a word taken into account?
What is necessary for this App to be used in other languages? How easy or difficult can this App be adapted to other languages?


**Paper Type:**

Demo

---

### Decision · Program_Chairs · 2023-03-17

Accept